# Effect of Titanium and Zirconia Nanoparticles on Human Gingival Mesenchymal Stromal Cells

**DOI:** 10.3390/ijms231710022

**Published:** 2022-09-02

**Authors:** Michael Nemec, Christian Behm, Vera Maierhofer, Jonas Gau, Anastasiya Kolba, Erwin Jonke, Xiaohui Rausch-Fan, Oleh Andrukhov

**Affiliations:** 1Clinical Division of Orthodontics, University Clinic of Dentistry, Medical University of Vienna, 1090 Vienna, Austria; 2Competence Center for Periodontal Research, University Clinic of Dentistry, Medical University of Vienna, 1090 Vienna, Austria; 3Clinical Division of Conservative Dentistry and Periodontology, University Clinic of Dentistry, Medical University of Vienna, 1090 Vienna, Austria; 4Center for Clinical Research, University Clinic of Dentistry, Medical University of Vienna, 1090 Vienna, Austria

**Keywords:** human gingival mesenchymal stromal cells, nanoparticles, dental implants, titanium, zirconia

## Abstract

Nano- and microparticles are currently being discussed as potential risk factors for peri-implant disease. In the present study, we compared the responses of human gingival mesenchymal stromal cells (hG-MSCs) on titanium and zirconia nanoparticles (<100 nm) in the absence and presence of *Porphyromonas gingivalis* lipopolysaccharide (LPS). The primary hG-MSCs were treated with titanium and zirconia nanoparticles in concentrations up to 2.000 µg/mL for 24 h, 72 h, and 168 h. Additionally, the cells were treated with different nanoparticles (25–100 µg/mL) in the presence of *P. gingivalis* LPS for 24 h. The cell proliferation and viability assay and live–dead and focal adhesion stainings were performed, and the expression levels of interleukin (IL)-6, IL-8, and monocyte chemoattractant protein (MCP)-1 were measured. The cell proliferation and viability were inhibited by the titanium (>1000 µg/mL) but not the zirconia nanoparticles, which was accompanied by enhanced apoptosis. Both types of nanoparticles (>25 µg/mL) induced the significant expression of IL-8 in gingival MSCs, and a slightly higher effect was observed for titanium nanoparticles. Both nanoparticles substantially enhanced the *P. gingivalis* LPS-induced IL-8 production; a higher effect was observed for zirconia nanoparticles. The production of inflammatory mediators by hG-MSCs is affected by the nanoparticles. This effect depends on the nanoparticle material and the presence of inflammatory stimuli.

## 1. Introduction

Dental implants have been proven to be secure and long-lasting options for replacing missing teeth, providing a pivotal treatment modality in modern dentistry [1,2]. The biocompatibility and favorable mechanical properties of titanium in dental implants led to the widespread implementation of this material in the emerging field of dental implantology [2]. However, since some adverse effects are associated with the titanium material, such as impaired aesthetics due to its dark color, as well as the patient demand for metal-free implants, the development of an alternative material has been promoted [3], and zirconia has appeared as a suitable alternative to titanium [4]. Besides their superior aesthetic properties, zirconia implants have shown decreased plaque accumulation on the surface and reduced inflammatory reaction rates in peri-implantitis tissues compared to titanium surfaces [4,5,6]. Further, zirconia implants exert a high level of biocompatibility [4], and the survival rates of zirconia implants are comparable to those of titanium ones [7,8].

Osseointegration is defined as the rigid fixation of an implant in the surrounding bone, and is pivotal for the clinical outcome and long-term stability [9]. Implants with moderately rough hydrophilic surfaces accelerate the osseointegration and allow earlier implant loading [10]. Regarding the implant material, zirconia implants show almost as good osseointegration as titanium ones [11]. Another crucial factor for the implant success is the integration of the dental implant into the surrounding soft tissue [12]. The quality of the soft tissue seal around the implant is considered inferior compared to the natural tooth and is more susceptible to various external factors and disease initiation [13]. The soft tissue integration is affected by various factors, such as the roughness, hydrophilicity, and coating with bioactive substances [14]. It is not clear to date how much the soft tissue healing and stability differ between Ti and Zr implants.

Although modern dental implants demonstrate a high success rate, some implant failures still occur. The major reason for implant loss is peri-implantitis, which is characterized by the inflammation of soft and hard peri-implant tissues [15]. The prevalence rates of peri-implantitis were estimated to range from 1 to 47% [16], which could be explained by differences in the definition and diagnosis of this disease. The etiology of peri-implantitis is unclear, but it is considered to be a multifactorial disease. Similarly to periodontitis, peri-implantitis is characterized by an increased bacterial load in the peri-implant pocket, although recent studies using 16s rRNA pyrosequencing showed some essential differences between periodontal and peri-implant microbiomes [17,18]. Moreover, the etiological role of the bacteria in peri-implantitis was also questioned, and an essential role of released micro- and nanoparticles in the initiation of immune response was suggested [19,20].

Material particles can be released during the implant placement [21] via corrosion over time [22] or due to therapeutic procedures such as implantoplasty [23]. These particles can diffuse up to 500 µm inside into peri-implant tissue. Particles of various sizes were detectable in tissues with peri-implantitis [24], and tissues affected with peri-implantitis showed higher particle concentrations [25]. The particle toxicity toward host cells depended on their amount and size, and the nanoparticles (NPs) exhibited higher toxicity than micron-sized particles [26]. In vitro studies have shown that NPs could be incorporated via endocytosis into cells [24,26], and might also have a genotoxic effect [27]. Due to the fact that the particles are incorporated but cannot be disintegrated by the organism, it is assumed that the particles could exert long-term effects on local cells [24]. However, the exact effect of the NPs on the cells of the peri-implant soft tissue is not entirely understood.

Human gingival mesenchymal stromal cells (hG-MSCs) are one of the major constituents of peri-implant soft tissue and play an important role in the production of the extracellular matrix and the inflammatory response [28,29]. Previous in vitro studies showed that the behavior and inflammatory characteristics of hG-MSCs are essentially influenced by the implant topography, hydrophilicity, and material [30,31,32]. Some previous studies suggested that the functional activity of hG-MSCs is influenced by Ti particles of different sizes [23,33,34]. One recent study showed a detrimental effect of both Ti and Zr particles with sizes of 60–100 nm and 2–75 µm, respectively, on the viability of a commercial human gingival fibroblast cell line [35]. However, the effect of the Ti and Zr nanoparticles of similar sizes on the primary hG-MSCs has never been investigated systematically. Therefore, in the present study, we compared the effects of Ti and Zr nanoparticles measuring <100 nm at various concentrations on the proliferation, viability, morphology, and inflammatory response of primary hG-MSCs. To mimic peri-implant conditions, hG-MSCs were stimulated with *Porphyromonas gingivalis* lipopolysaccharide (LPS), because this putative periodontal pathogen is often found in peri-implant lesions [36].

## 2. Results

### 2.1. Cell Proliferation and Viability

The proliferation and viability rates of the hG-MSCs after the treatment with Ti and Tz NPs for 24 h, 72 h, and 168 h are presented in Figure 1. Throughout the experiment, the Ti NPs inhibited the hG-MSCs’ proliferation and viability in a concentration-dependent manner; a statistically significant inhibitory effect was observed starting from the concentration of 1000 µg/mL. In contrast, the Zr NPs had no inhibitory effect on the hG-MSCs’ proliferation or viability at all investigated concentrations and time points. Moreover, in some cases, a significant increase in cell proliferation or viability after treatment with Zr NPs was observed.

### 2.2. Live–Dead Staining

The live–dead stainings of hG-MSCs cultured in the presence or in the absence of different NPs at a concentration of 1000 µg/mL are presented in Figure 2. In the absence of NPs, the majority of cells were viable and not stained with PI. In the presence of NPs, a markedly higher number of dead PI-positive cells was observed. The number of PI-positive cells was visually higher after the treatment with Ti NPs compared to Zr NPs at all time points.

### 2.3. Focal Adhesion Staining

Figure 3 shows the focal adhesion staining of hG-MSCs grown in the presence of Ti or Zr NPs at a concentration of 250 µg/mL, as well hG-MSCs growing without NPs after 24 h, 72 h, and 168 h of culture. The hG-MSCs grown in the absence of NPs had a classic spindle-like shape throughout the whole observation period. Both types of NPs attached to the surfaces of the hG-MSCs, and this resulted in changes in their morphology; in particular, they became less prolonged compared to the control group. No visible differences between cells cultured in the presence of Ti NPs and Zr NPs were observed.

### 2.4. Effect of NPs on the Basal Production of IL-6, IL-8, and MCP-1 in hG-MSCs

The effects of different NPs at concentrations of 250 and 1000 µg/mL on the basal production of IL-6, IL-8, and MCP-1 in hG-MSCs after 24 h, 72 h, and 168 h are shown in Figure 4, Figure 5 and Figure 6, respectively. The gene expression of IL-6 was not affected by either type of NP after 24 h and decreased by 1000 ng/mL Zr NPs after 72 h and 168 h, or by either type of NP at a concentration of 250 µg/mL after 168 h. The production of the IL-6 protein was slightly enhanced by 1000 µg/mL of Zr NPs after 24 h, whereas no significant effect of NPs on the IL-6 protein levels was observed under any other condition tested.

Both types of NPs significantly enhanced the gene expression of IL-8 at all concentrations throughout the whole observation period (Figure 5). After 72 h and 168 h, a significantly higher IL-8 protein production rate upon the incubation with NPs was observed compared to the control group. After 72 h, the hG-MSCs treated with Ti particles exhibited significantly higher IL-8 gene expression levels compared to those treated with Zr NPs at similar concentrations. However, at the protein level, these data were confirmed only at the NP concentration of 250 µg/mL, whereas at the concentration of 1000 µg/mL an opposite effect was observed. Significantly higher IL-8 protein production was observed.

No significant effect of any NP on the gene or protein expression levels of MCP-1 was observed (Figure 6). After 72 h and 168 h, the MCP-1 gene expression after the treatment with Ti NPs at a concentration of 1000 µg/mL was significantly higher compared with the Zr NPs at a similar concentration, but this was not observed at the protein level. In addition, the production of the MCP-1 protein after the treatment with Zr NPs at a concentration of 250 µg/mL was slightly but significantly higher compared to the Ti NPs.

### 2.5. Effect of NPs on the P. gingivalis LPS-Induced Production of IL-6, IL-8, and MCP-1 in hG-MSCs

The effects of various NPs on the *P. gingivalis* LPS-induced response in hG-MSCs after 24 h of incubation are shown in Figure 7. The *P. gingivalis* LPS significantly increased the gene expression levels and protein production rates of IL-6, IL-8, and MCP-1. The NPs did not affect the *P. gingivalis* LPS-induced IL-6 gene expression or protein production. The *P. gingivalis* LPS-induced IL-8 gene expression was significantly enhanced by the Zr NPs at a concentration of 100 µg/mL, and the IL-8 protein production was enhanced by the Zr NPs at concentrations of 25 and 100 µg/mL and by Ti NPs at a concentration of 100 µg/mL. Additionally, in the presence of *P. gingivalis* LPS, the Zr NPs induced significantly higher IL-8 protein production rates than Ti NPs at the NP concentration of 100 µg/mL. The *P. gingivalis* LPS-induced MCP-1 gene expression was significantly inhibited by the NPs, except for the Ti NPs at a concentration of 25 µg/mL. Under these conditions, the gene expression of MCP-1 in the presence of the Zr NPs was significantly lower than those in the presence of the Ti NPs. No significant effect of the NPs on the MCP-1 protein production in the presence of *P. gingivalis* LPS was observed.

## 3. Discussion

In the present study, we focused on a comparison of the effects of Zr and Ti NPs on primary hG-MSCs. Our data showed that Ti particles exhibit higher cytotoxicity toward hG-MSCs than Zr particles. This was suggested by the finding that the proliferation or viability of hG-MSCs was inhibited by the Ti particles starting from the concentration of 1000 µg/mL, whereas no cytotoxic effect of Zr particles was observed. A higher cytotoxicity of Ti NPs was also confirmed by the live–dead staining. These data are in agreement with the previous data with particles of a similar size. In particular, the cytotoxicity of the Ti nanoparticles in immortalized human periodontal ligament cells (EC50 2.8 mg/mL) was up to five times higher than that of Zr nanoparticles (EC50 13.96 mg/mL) [26]. The reasons for the difference in the toxicity between Ti and Zr particles are not clear. One recent study showed that the cytotoxicity of Ti nanoparticles was determined by the shape of the particles rather than by their small size [37]. Ti and Zr particles could have different forms, which could explain the different effects on the hG-MSCs’ proliferation and viability.

The analysis of the cell morphology showed that the presence of nanoparticles resulted in an altered cell morphology. The presence of nanoparticles or their conglomerates provided some additional attachment places for the cells, leading to the different morphologies. This assumption was confirmed by the re-distribution of vinculin staining in the presence of nanoparticles (Figure 3). Under this condition, vinculin was localized in the area of nanoparticles, suggesting the formation of focal adhesion complexes. The alteration of the cell shape can result in different functional properties. Particularly, one study showed that the cell shape might determine the differentiation fate of human mesenchymal stem cells independently of other parameters [38]. Thus, the material particles in the tissue can serve as an attachment point for the resident cells and can alter their function.

We further investigated the effects of various nanoparticles on the inflammatory response of the hG-MSCs. This was done in the presence and in the absence of *P. gingivalis* LPS. In the absence of *P. gingivalis* LPS, only rather high concentrations of nanoparticles (>250 ng/mL) were able to upregulate the production of pro-inflammatory mediators via hG-MSCs. Some differences were observed between the nanoparticles of different materials. Interestingly, at the gene level, in many cases the expression level of pro-inflammatory mediators in hG-MSCs was higher after the treatment with Ti compared to the zirconia nanoparticles. However, these findings were rarely confirmed by the protein data, and even an opposite tendency was observed. Thus, the Zr nanoparticles induced significantly higher IL-6 and IL-8 protein production levels after 24 h and greater MCP-1 production after 72 h than Ti NPs, despite the similar gene expression levels. Moreover, 1000 µg/mL of Zr NPs induced a significantly higher IL-8 protein concentration than the Ti NPs, although at the gene level an opposite effect was observed. The lacking correspondence between the gene expression and protein data could be partially due to the cytotoxic effect of the Ti NPs, which can lead to a lower number of functionally active cells and to lower protein production.

Previous studies differently described the ability of material particles to induce the inflammatory response in different host cells. In gingival fibroblasts, a low concentration of Ti particles (in the range of µg/mL) induced the production of IL-6 [33]. In fibroblasts isolated from peri-implant granulation tissue, Ti particles with an average size of 2.5 µm induced the production of TNF-α, IL-6, and IL-8 [34]. Ti particles with a submicron size (0.1–0.4 µm) induced the production of IL-1β, TNF-α, and IL-6 in THP-1 macrophages [39]. Ti particles measuring several micrometers induced the production of IL-1β, IL-6, and TNF-α in primary mice macrophages [40]. In contrast, a recent study did not observe any upregulation of IL-1β or IL-6 release by THP-1 cells upon stimulation with Ti particles with sizes of 60–100 nm and with Zr particles with sizes of 2–75 µm [35].

The material NPs also modified the response of hG-MSCs to *P. gingivalis* LPS, which was used to mimic the inflammatory environment characteristics for the peri-implantitis. We found that the NP-modified *P. gingivalis* induced an inflammatory response depending on the cytokine type and NP material. Thus, both types of NPs enhanced the *P. gingivalis* LPS-induced Il-8 production, and this effect was slightly higher for Zr NPs than Ti NPs. In contrast, the MCP-1 expression was reduced by both NPs, and higher inhibition was observed for Zr than Ti NPs. IL-8 is a potent chemokine for polymorphonuclear neutrophils, and MCP-1 stimulates monocyte migration. A stimulating effect of Ti particles measuring over 0.22 µm on LPS-induced IL-1β production was previously reported in THP-1 macrophages [41]. The translation of these data to the clinical situation implied that the presence of NPs modifies the infiltration of immune cells in the peri-implant tissue and influences the immune response.

In our study, we focused on the particles with a size of about 100 nm. The particles of this size are released during implantoplasty [23], which is a common treatment option for patients with peri-implantitis. However, NPs are only one fraction of the particles observed in the tissues around the implant. The size of the material particles in the peri-implant tissue is estimated to be in the range from nanometers up to 20 µm [42]. There is some evidence that nanoparticles could be more toxic than microparticles [43], which could be due to the ability of host cells to phagocyte nanoparticles [44]. However, to estimate the effects of different material particles, studies with different particle sizes would be necessary.

The role of material particles in the etiology of peri-implantitis is debatable to date [17,45,46]. The recent data suggest several dissimilarities in the etiologies of periodontitis and peri-implantitis and a potential role of a particle-induced response in this process [17]. Several studies observed Ti particles in peri-implant soft tissue [47,48,49]. An in vivo study observed Ti ions at distances up to 1 mm from the implant, and the amount of Ti ions did not depend on the implant roughness [50]. The concentration of the Ti is substantially increased in peri-implantitis compared to the control [48]. One histological study showed that TP also induces an immunological reaction [48], whereas another study denied this association [49]. Nevertheless, the presence of particles in peri-implant tissue might be an important factor influencing the inflammatory processes.

Zirconia seems to be superior to Ti in the terms of the release of particles and their adverse effects in gingival tissue. Our data and other studies show that the toxic effect of material particles is observed for Zr at higher concentrations than for Ti. Furthermore, zirconia as a non-metal is assumed to release fewer particles than titanium. This statement is supported by one study showing that platform-switched constructs consisting of a Ti implant and Zr abutment released up to 10 times less particles than platform-matched constructs consisting of Ti upon the application of static forces and an acidic environment [51]. A recent study on a mini pig found about a three-fold higher content of Ti in the bone adjacent to the Ti implant than that of Zr in the bone around the Zr implant [26]. However, it should be noted that the size of particles released could also depend on the material. In particular, Ti was reported to release particles measuring > 1 µm, whereas for Zr the particles were > 1 µm [26,43]. Thus, the concentration of nanoparticles around a Zr implant could be higher than that around a Ti implant.

Some of the effects of NPs in our study were observed at physiologically relevant concentrations. There have been different estimations about the amount of material particles in peri-implant tissue. Thus, a study on a mini pig reported up to 2.17 mg/kg of bone Ti and 0.59 mg/kg of bone zirconia around Ti and Zr implants placed in the maxilla [26], which corresponded to concentrations of material particles of 2.17 µg/mL and 0.59 µg/mL, respectively. Another study on humans reported that the concentration of Ti in the bone near the implant could reach 38 mg/kg bone [43], which corresponded to 38 µg/mL of Ti particles. These data suggest that the content of material particles in the peri-implant tissue is high enough to modify the cell response to the bacterial component and could be clinically relevant. However, these concentrations are still substantially lower than 1000 µg/mL, the concentration of Ti NPs at which the inhibitory effect on the hG-MSC viability was observed. It cannot be excluded that the local concentration in some areas is even higher, and some toxic effects of Ti particles can still take place in vivo.

The main limitation of the present study is the in vitro design. We used the cells isolated from systemically and periodontally healthy young individuals, and this population does not reflect peri-implantitis patients. We used only one cells type, whereas the inflammatory response in peri-implantitis involves various cells. In our study, we investigated only nanoparticles, but implant release particles of different sizes. Although NPs exhibit the most toxic effects on the host cells, the contribution of microparticles to the host response should be considered. Finally, through the interpretation of the effects of Ti and Zr, the differences in release rate and size between these materials should be considered.

## 4. Materials and Methods

### 4.1. Ethical Consideration

Ethical approval for the hG-MSC isolation was obtained from the Ethics Committee of the Medical University of Vienna, Vienna, Austria (vote no.1079_2019). Written and informed patient consent was given prior to tooth donations. The present study was conducted in compliance with the Declaration of Helsinki regarding the ethical principles for medical research involving human subjects and according to the Good Scientific Practice Guidelines of the Medical University of Vienna.

### 4.2. Cell Isolation

Primary hG-MSCs were isolated from the third molar teeth of periodontally healthy individuals aged between 18 and 32 years, which were extracted due to orthodontic indications. The donors were periodontally and systemically healthy and did not take any regular medication three months prior to the study. The cells were isolated using the outgrowth method [52,53]. The isolated cells were cultured in Dulbecco’s modified Eagle´s medium (DMEM; Sigma-Aldrich, St. Louis, MO, USA) supplemented with 10% fetal bovine serum (FBS, Gibco, Carlsbad, CA, USA) and 100 U/mL penicillin/100 µg/mL streptomycin (Pen-Strep, Gibco, Carlsbad, USA) at 37 °C, 5% CO_2_, and 95% humidity. The hG-MSCs were passaged after reaching 80% to 90% confluence and the medium was changed every three days. The cells between passages four and six were used for all experiments.

### 4.3. Nanoparticles

Two different commercially available (Sigma-Aldrich, St. Louis, USA) nanoparticles, titanium (IV)dioxide (TiO_2_, Ti NPs) nanopowder and zirconium (IV) dioxide (ZrO_2_, Zr NPs) nanopowder, each with a particle size of <100 nm, were used. The particles were suspended in DMEM, supplemented with Pen-Strep without any FBS, to obtain a 2000 µg/mL stock solution.

### 4.4. Cell Seeding and Stimulation

Here, 5 × 10^4^ hG-MSCs were seeded per well in 24-well plates using 500 µL DMEM supplemented with 10% FBS and Pen-Strep. After 24 h of incubation, the hG-MSCs were stimulated with different concentrations of either TiO2 or ZrO2. In some experiments, the stimulation was performed in the presence of 1 µg/mL standard *P. gingivalis* LPS (Invivogen, San Diego, CA, USA) and 200 ng/mL soluble CD14 (sCD14, Sigma-Aldrich, St. Louis, USA) [54]. For the stimulation, the medium was changed to FBS-free DMEM supplemented with Pen-Strep. The hG-MSCs treated with FBS-free DMEM served as the control.

### 4.5. Cell Proliferation and Viability

The cell viability and proliferation were evaluated after 24 h, 72 h, and 168 h of stimulation using 3-(4,5-dimethylthiazol-2-yl)-2,5-diphenyltetrazoliumbromid (MTT, Sigma Aldrich, St. Louis, MI, USA) as described previously [55]. In brief, 100 µL MTT solution (5 mg/mL in 1× phosphate-buffered solution (PBS)) was added per well, followed by incubation at 37 °C for 2 h. After discarding the conditioned media, 500 µL dimethyl sulfoxide (Merck, Darmstadt, Germany) was added per well, followed by 5 min incubation. To reduce the risk of optical interference due to the nanoparticles, the resulting solutions were transferred into 5 mL tubes and centrifuged. Finally, 100 µL of the supernatant was transferred in quadruplicate to a 96-well plate, and the optical density (OD) was measured at 570 nm with a photometer (Synergy HTX, Biotek, Winooski, VT, USA).

### 4.6. Live–Dead Staining

The live–dead analysis was performed after 24 h, 72 h, and 168 h of incubation using the Live/Dead Cell Staining Kit from Enzo Life Sciences (Lausen, Switzerland) following the manufacturer’s guidelines. Briefly, after washing the cells with 1 × PBS, 150 µL of staining solution containing calcein-AM and propidium iodide for staining live and dead cells, respectively, was added per well followed by incubation for 15 min at 37 °C, 5% CO_2_, and 95% humidity. Afterwards, the fluorescence staining was directly visualized using the EchoRevolve fluorescence microscope (Echo, San Diego, CA, USA), distinguishing between calcein-AM (green dye, Ex/Em = 488/515 nm) and propidium iodide (Ex/Em = 570/602 nm).

### 4.7. Focal Adhesion Staining

The focal adhesions were stained after 24 h, 72 h, and 168 h of incubation for hG-MSCs using the Actin Cytoskeleton/Focal Adhesion Staining Kit from Merck (Darmstadt, Germany), according to the manufacturer’s instructions. In brief, after fixing and permeabilizing the cells with 4% paraformaldehyde and 0.1% Trition X-100 in 1 × PBS, respectively, the hG-MSCs were blocked by adding 1% BSA in 1 × PBS. The actin filaments were stained with TRITC-conjugated anti-phalloidin (Ex/Em 540/565 nM) for focal adhesion with the anti-vinculin monoclonal antibody (purified clone 7F9) followed by the FITC-conjugated secondary antibody (Ex/Em 495/520 nm). This was followed by nuclei counterstaining with 4′,6-diamidin-2-phenylindol (DAPI, Ex/Em 358/461 nm). Brightfield and fluorescence pictures were taken using the EchoRevolve fluorescence microscope (Echo, San Diego, CA, USA).

### 4.8. Production of Inflammatory Mediators by hG-MSCs

The effects of different nanoparticles on the production of inflammatory mediators by hG-MSCs was assessed in the absence or in the presence of *P. gingivalis* LPS based on the production of the pro-inflammatory mediators IL-6, IL-8, and MCP-1. In the absence of *P. gingivalis* LPS, the inflammatory response was evaluated after 24 h, 72 h, and 168 h, whereas in the presence of LPS it was done after 24 h.

The gene expression levels were quantified using qPCR, similarly to our previously published study [56]. Cell lysis, mRNA transcription into cDNA, and qPCR processes were performed using the TaqMan^®^ Gene Expression Cells-to-CT™ kit (Ambion/Applied Biosystems, Foster City, CA, USA) according to the manufacturer’s instructions. The reverse transcription was performed using a Primus 96 advanced thermocycler (Applied Biosystems, Foster City, CA, USA) by heating the samples to 37 °C for one hour followed by 95 °C for five minutes. Here, qPCR was performed on an ABI StepOnePlus device (Applied Biosystems, Foster City, CA, USA) in paired reactions using TaqMan^®^ gene expression assays with the following ID numbers (all from Applied Biosystems): interleukin-8 (IL-8) (Hs00174103_m1), interleukin-6 (IL-6) (Hs00985639_m1), monocyte chemoattractant protein (MCP-1) (Hs00234140_m1); glyceraldehyde 3-phosphate dehydrogenase (GAPDH) (Hs99999905_m1). GAPDH was used as a house-keeping gene. The PCR reactions were performed in triplicate under the following conditions: 95 °C for 10 min, followed by 50 cycles, each for 15 s at 95 °C and at 60 °C for 1 min. For each sample, the point at which the PCR product was first detected above a fixed threshold (cycle threshold, Ct) was determined. The 2^−ΔΔCt^ method was used to calculate the relative expression of target genes by taking unstimulated hG-MSCs as a control; here, ΔΔCt was calculated using the following formula:$$∆∆C_{t}={\left(C_{t}{}^{target}-C_{t}{}^{GAPDH}\right)}_{sample}-{\left(C_{t}{}^{target}-C_{t}{}^{GAPDH}\right)}_{control}$$

The levels of IL-6, IL-8, and MCP-1 proteins in conditioned media were determined using a commercially available ELISA (ThermoFisher Scientific, Waltham, MA, USA) according to the manufacturer’s instructions. The conditioned media were harvested and centrifuged to remove the cell debris and nanoparticles. For all ELISAs, the OD values were measured at 450 nm and 570 nm using a Synergy HTX multi-mode reader (BioTek Instruments, Winooski, VT, USA) and their concentrations in the conditioned media were calculated based on the standard curves ranging 3.125–200 pg/mL for IL-6, 3.9–250 pg/mL for IL-8, and 15.6–1000 pg/mL for MCP-1. All values below the detection limit were considered as zero for the analysis.

### 4.9. Statistical Analysis

All experiments were repeated at least five times, with cells isolated from five different donors. Each experiment was performed at least in technical triplicate. Due to the low number of experiments, it was not possible to confirm the normal distribution, meaning non-parametric tests were used. The differences between groups were assessed using the Wilcoxon signed-rank test. Here, *p* values less than 0.05 were considered to be statistically significant. The data are presented as means ± standard errors of the mean. All statistical analyses were performed using SPSS 24.0 software (IBM, Armonk, NY, USA). The evaluation of LDS and FAK images was performed qualitatively.

## 5. Conclusions

The NPs released from the implant material influence the functional activity of the primary hG-MSCs under normal and inflammatory conditions. Our data show that at similar concentrations, the Ti NPs have a more detrimental effect on the hG-MSCs compared to the Zr NPs. Taking into consideration the generally lower release of particles by Zr than Ti, our data suggest that Zr implants could be superior in terms of the potential side effects due to particle release.

## Figures and Tables

**Figure 1 ijms-23-10022-f001:**
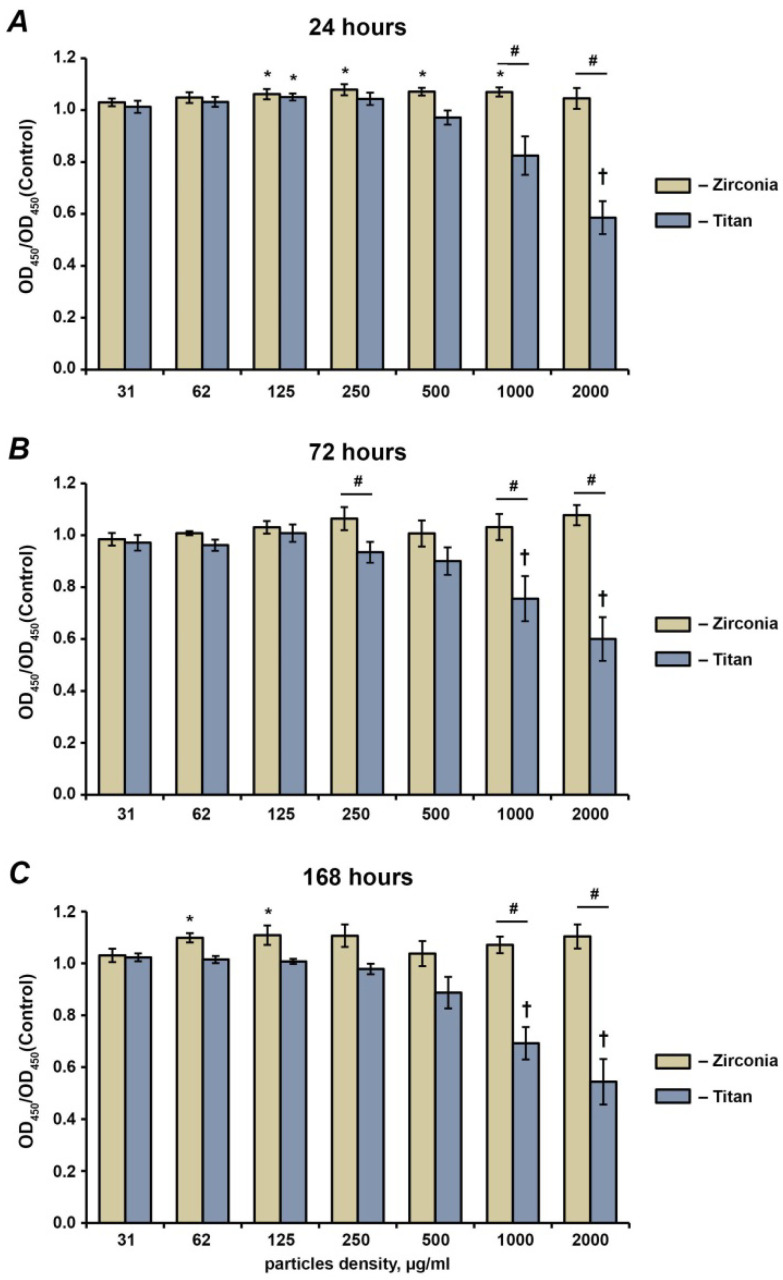
Effect of different NPs on the proliferation and viability of hG-MSCs. The hG-MSCs were cultured in the absence or in the presence of titanium or zirconia NPs at different concentrations for 24 h (**A**), 72 h (**B**), or 168 h (**C**). The proliferation and viability were measured using the MTT method. The Y-axis represents the ratios of optical densities (OD) measured at 570 nm in hG-MSCs cultured with NPs to those in the control group (without NPs). Data are presented as means ± s.e.m. of six independent experiments with hG-MSCs isolated from six different donors. Note: *—significantly higher compared to control, *p* < 0.05; †—significantly lower compared to control, *p* < 0.05; #—significantly different between Ti NPs and Zr NPs, *p* < 0.05.

**Figure 2 ijms-23-10022-f002:**
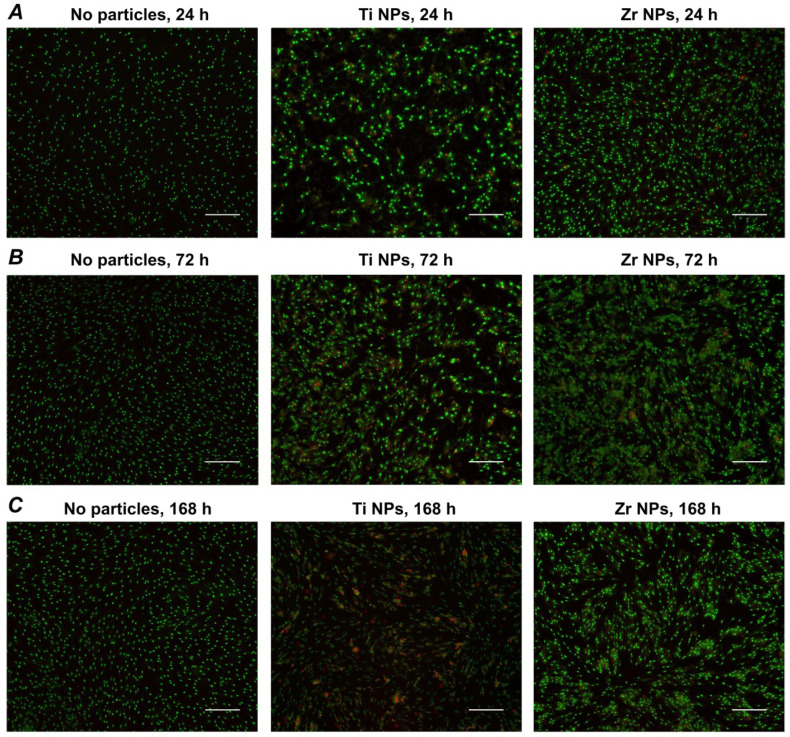
Live–dead staining of hG-MSCs cultured with or without different NPs. The hG-MSCs were cultured in the absence or in the presence of Ti NPs or Zr NPs at a concentration of 1000 µg/mL for 24 h (**A**), 72 h (**B**), or 168 h (**C**) and stained with a live–dead staining kit. Viable and dead cells are visualized in green and red, respectively. Images were taken from a representative experiment. The scale bar corresponds to 200 µm.

**Figure 3 ijms-23-10022-f003:**
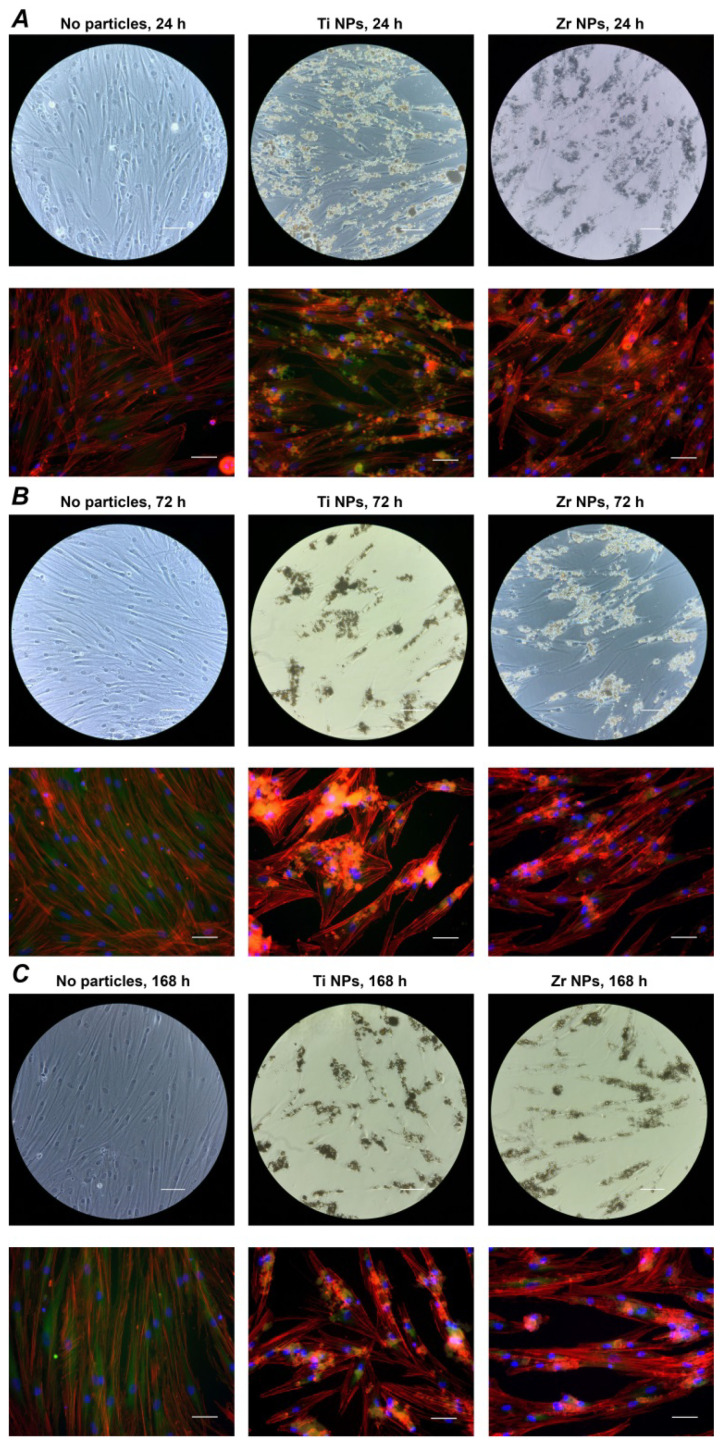
Focal adhesion staining of hG-MSCs cultured with or without NPs. The hG-MSCs were cultured in the absence or in the presence of Ti NPs or Zr NPs at a concentration of 250 µg/mL for 24 h (**A**), 72 h (**B**), or 168 h (**C**) and the cells were stained with a focal adhesion staining kit. F-actin was stained with TRITC-conjugated phalloidin (red), the focal adhesions with anti-vinculin and FITC-conjugated secondary antibody (green), and the nucleus with DAPI (blue). The scale bars correspond to 20 µm.

**Figure 4 ijms-23-10022-f004:**
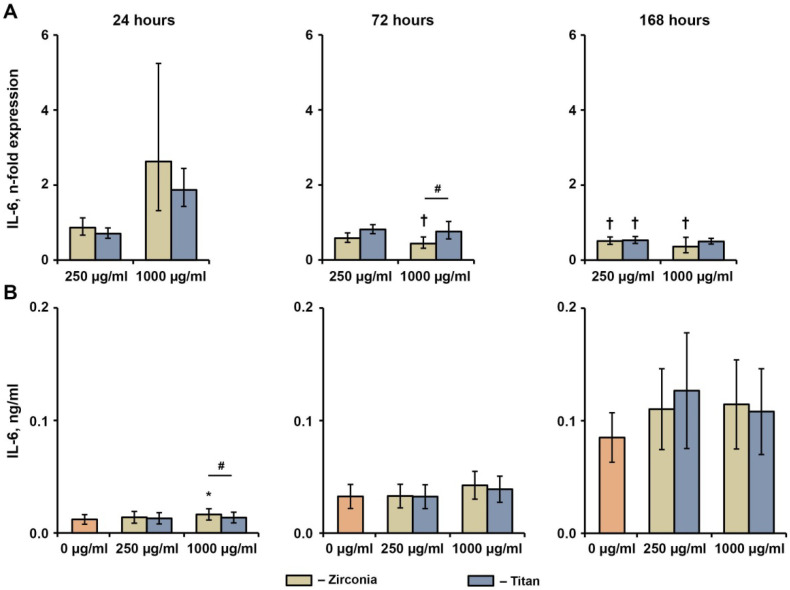
Effects of NPs on the basal IL-6 expression in hG-MSCs. The hG-MSCs were cultured in the absence or in the presence of different NPs for 24 h, 72 h, or 168 h. The resulting IL-6 gene expression (**A**) and IL-6 protein production (**B**) rates were measured via qPCR and ELISA, respectively. (**A**) Y-axis represents n-fold expression of IL-6 in hG-MSCs cultured with NPs compared to the unstimulated control (n-fold expression = 1). N-fold expression was calculated using the 2^−ΔΔCt^ method, using GAPDH as the reference gene. (**B**) Y-axis shows the concentration of IL-6 in the conditioned media. Data are presented as means ± s.e.m. of six independent experiments performed with hG-MSCs isolated from six different donors. Note: *—significantly higher compared to control, *p* < 0.05; †—significantly lower compared to control, *p* < 0.05; #—significantly different between Ti NPs and Zr NPs, *p* < 0.05.

**Figure 5 ijms-23-10022-f005:**
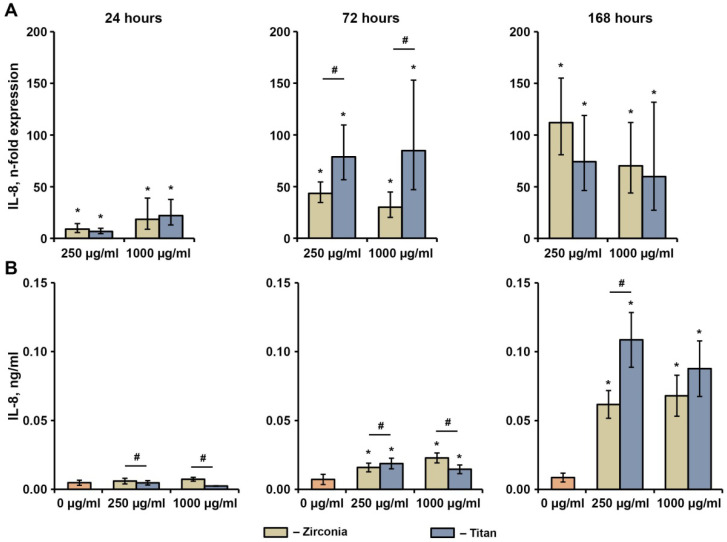
Effects of NPs on the basal IL-8 expression in hG-MSCs. The hG-MSCs were cultured in the absence or in the presence of different NPs for 24 h, 72 h, or 168 h. The resulting IL-8 gene expression (**A**) and IL-8 protein production (**B**) rates were measured via qPCR and ELISA, respectively. (**A**) Y-axis represents n-fold expression of IL-8 in hG-MSCs cultured with NPs compared to unstimulated control (n-fold expression = 1). N-fold expression was calculated using the 2^−ΔΔCt^ method, using GAPDH as the reference gene. (**B**) Y-axis shows the concentration of IL-8 in the conditioned media Data are presented as means ± s.e.m. of six independent experiments performed with hG-MSCs isolated from six different donors. Note: *—significantly higher compared to control, *p* < 0.05; #—significantly different between Ti NPs and Zr NPs, *p* < 0.05.

**Figure 6 ijms-23-10022-f006:**
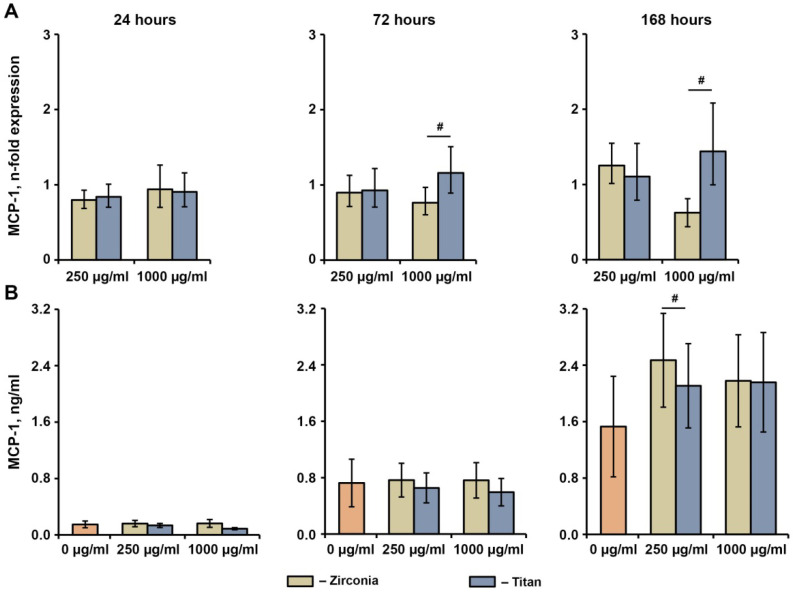
Effects of NPs on the basal MCP-1 expression in hG-MSCs. The hG-MSCs were cultured in the absence or in the presence of different NPs for 24 h, 72 h, or 168 h. The resulting MCP-1 gene expression (**A**) and MCP-1 protein production (**B**) rates were measured via qPCR and ELISA, respectively. (**A**) Y-axis represents n-fold expression of MCP-1 in hG-MSCs cultured with NPs compared to unstimulated control (n-fold expression = 1). N-fold expression was calculated using the 2^−ΔΔCt^ method, using GAPDH as the reference gene. (**B**) Y-axis shows the concentration of MCP-1 in the conditioned media Data are presented as means ± s.e.m. of six independent experiments performed with hG-MSCs isolated from six different donors. Note: #—significantly different between Ti NPs and Zr NPs, *p* < 0.05.

**Figure 7 ijms-23-10022-f007:**
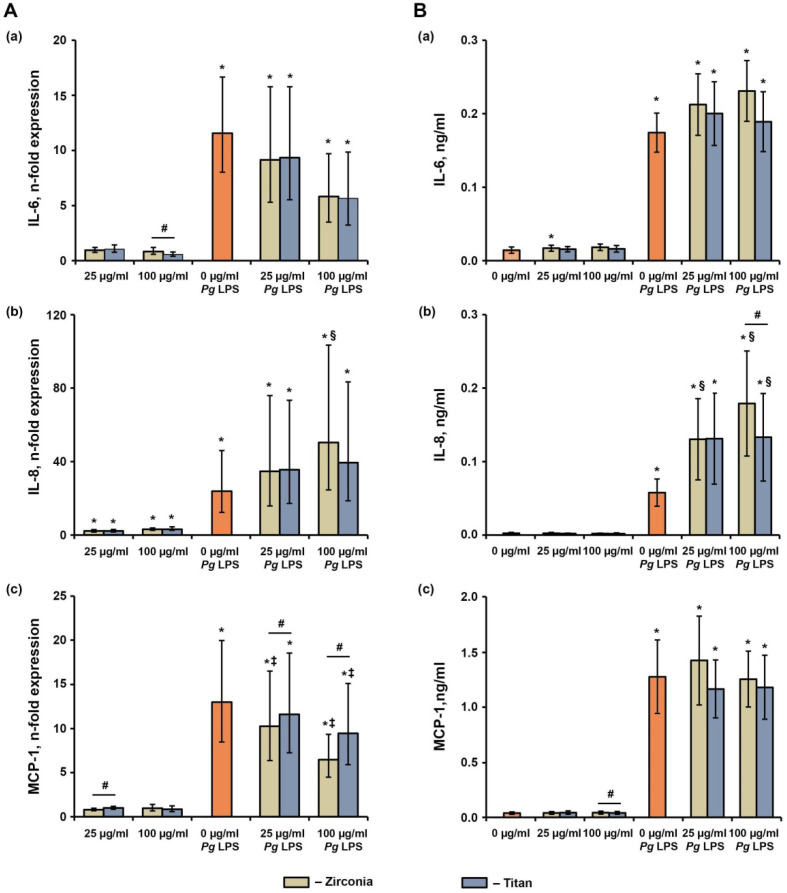
The effects of different NPs on the *P. gingivalis* LPS-induced response in hG-MSCs. The hG-MSCs were stimulated with *P. gingivalis* LPS (Pg LPS) in the absence or in the presence of different NPs for 24 h. The resulting gene expression (**A**) and protein production (**B**) rates for IL-6, IL-8, and MCP-1 were determined via qPCR and ELISA, respectively. (**A**) Y-axis represents n-fold expression of target genes in hG-MSCs cultured under various conditions in relation to cells cultured without NPs and Pg LPS, calculated using the 2^−ΔΔCt^ method, using GAPDH as a reference gene. (**B**) Y-axis shows the concentrations of various proteins in the conditioned media measured. Data are presented as means ± s.e.m. of six independent experiments performed with hG-MSCs isolated from six different donors. Note: *—significantly higher compared to control, *p* < 0.05; #—significantly different between Ti NPs and Zr NPs, *p* < 0.05; §—significantly higher compared to Pg LPS without NPs, *p* < 0.05; ‡—significantly lower compared to Pg LPS without NPs, *p* < 0.05.

## Data Availability

The data presented in this study are available on request from the corresponding author.

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
