# Peer review of "Effect of Titanium and Zirconia Nanoparticles on Human Gingival Mesenchymal Stromal Cells"

_ijms, 2022, doi:10.3390/ijms231710022_

Round 1

Reviewer 1 Report

In my opinion, it is a very interesting paper. The study design is appropriate.

The aim of this study should be clearly mentioned in the abstract and the introduction sections. 

The methodology was well presented. 

Statistical analysis has been adequately described. All figures  (histograms and images)  helped to clearly present the results. 

Ethical issues have been adequately addressed. The conclusions are specific and clear.

Author Response

We are thankful for positive evaluation of our manuscript.

Reviewer 2 Report

A particularly interesting article, with conclusions of practical importance

Author Response

(The authors gave the same response as above.)

Reviewer 3 Report

The study by the authors was very well-designed.

They studied the impact of the nanoparticles of two main types of implants currently being used for implant treatment i.e. titanium and zirconium, on soft tissue. This study further clarifies one of the factors causing periimplantitis. For that they applied different concentrations of particles and assessed their effect on the gingival mesenchymal cells in presence and absence of P. gingivalis, an inflammatory mediator often released in peri implantitis. The novelty related to the research is the first time reporting of the impact of similar sized particles with different concentrations on gingival cells. This was confirmed by assessing cytokines levels at different follow-up timepoints. The found that zirconia based implants had less detrimental effect on the soft tissue cells compared to titanium.

The findings of the current study could be potentially beneficial for clinicians and readers as it further strengthens the fact that zirconia implants are more safer than conventional titanium implants.  Furthermore, it provides evidence related to the etiology of peri-implantitis. The presented data could allow a clinician to avoid the risk of peri-implantitis by incorporating zirconia implants in a clinical practice.

Author Response

(The authors gave the same response as above.)
